# An Investigation of Thermal Effects on Micro-Properties of Sudbury Norite by CT Scanning and Image Processing Method

Sheng-Lin Wang [1,*], Brad Simser [2], Shunde Yin [1] and Ju Huyan [3]

[1] Department of Civil and Environmental Engineering, Faculty of Engineering, University of Waterloo, Waterloo, ON N2L 3G1, Canada
[2] Sudbury Integrated Nickel Operations, Glencore, Sudbury, ON P0M 1S0, Canada
[3] School of Transportation, Southeast University, Nanjing 210000, China
[*] Correspondence: s545wang@uwaterloo.ca

**Abstract:** Rock is constantly subjected to stress and thermal conditions. Thermal-induced micro-cracks will be generated as a result of different thermal expansion gradations between different minerals. This characteristic was investigated in this paper by studying the micro-properties of Sudbury norite via CT scanning and the image processing method. A novel filtering method, maximum–minimum shadow filtering (MMSF), was developed in this study to highlight the thermal-induced micro-cracks in Sudbury Igneous Complex (SIC) norite after different temperature treatments. Based on quantitative analysis, the areal percentages of biotite, felspar, quartz, and small amounts of metal minerals were determined. It was also found that small-scale micro-cracks were first observed in the middle of biotite grains at a temperature of 400 °C. The cracks further propagated and extended with the temperature increase. In addition, the orientations of cracks either remained at the same distribution or became more evenly distributed with the rising temperature. A linear relationship was found between the average porosity of SIC norite and the temperature. Moreover, the anisotropic properties between vertical and horizontal directions of norite were also noticeable. Overall, the paper presented a quantitative study on the effects of thermal treatment and the anisotropic properties of SIC norite. Methodology and findings from this paper will be a significant reference for future studies regarding the thermal impacts on norite and similar rocks.

**Keywords:** Sudbury norite; thermal effect; image processing; quantitative analysis; micro-crack

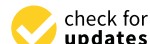



## 1. Introduction

Rock, as a collection of minerals, is constantly subjected to stress and thermal conditions [1–4]. Different minerals will have different thermal expansion rates under temperature rising, thus causing thermal stress under high temperatures. During the process of crustal movement and mankind's activities, such as drilling, mining, and earthquakes, rocks may expose to higher-than-normal temperatures [5]. Such thermal and exhumation effects could induce the appearance of micro-cracks which were reported in granite and sandstone especially when the condition temperature exceeds 400 °C [1,6,7]. Based on recent studies, three major thermal damage stages were indicated [8,9]: (1) <400 °C, a release of free water, crystal and structural water causes micro-cracks and voids to emerge; (2) 400–600 °C, the mineral composition starts to change and the transition of quartz from $\alpha$ phase to the $\beta$ phase occurs, increasing micro-cracks, (3) 600–1000 °C, severe damage of rock mass occurs due to mineral expansion, decomposition, metallic bond breaking, and melting. Recently, with developed data visualization methods, more quantitative analysis, and Computed Tomography (CT) scanning were conducted. Table 1 summarizes the recent methodology and key findings for evaluating the thermal effects on several types of rocks.

**Table 1.** Recent literature on the thermal effects of rocks.

| Recourse | Rock Type | Heating Method | Key Findings |
|---|---|---|---|
| [6] | Sandstone | 25–600 °C Rate: 5 °C/min Condition: 2 h | Between 400 and 600 °C, the pore volume, porosity and peak strain increased rapidly, while the strength decreases rapidly. |
| [1] | Granite | 25–800 °C Rate: 2.5 °C/min Condition: 6 h | The heterogeneity coefficient reached its peak at around 500 °C. The anisotropy coefficient remained steady from 20 °C to 500 °C. It then decreased sharply after 600 °C. |
| [10] | Granite | 100–1000 °C Rate: 5 °C/min Condition: 24 h | The pore network models (PNMs) of thermally treated rock were developed and enabled the quantifications of the size and distribution of the pores in granite. |
| [7] | Shale rock | 25–500 °C Rate: 1 °C/min Condition: 24 h | P-wave velocity and brittleness index of shale decreased due to dehydration and organic matter burning. In addition, the temperature rising made the shale more homogeneous and less anisotropic. |
| [8] | Red sandstone | 10–1300 °C Rate: 10 °C/min Condition: 1 h | The porosity of sandstone increased at temperatures starting from 500 °C and peaking at 1000 °C. |

Norites are found in a variety of environments on earth, ranging from oceanic crust [11] to layered intrusions [12], and found to impact melt sheets, such as the Sudbury Igneous Complex (SIC) [13,14]. In particular, SIC norite contains high absolute rare-earth element (REE) contents and is usually observed in Nickel (Ni) and Copper (Cu) ores. Therefore, research into the SIC norite holds both scientific and economic importance. Past literature mainly focused on its origins, chemical evolutions, and REE characterizations, and is dated. In contrast, studies on microstructure, thermal resistance property and crack propagation characteristics of norite are rarely seen.

In this regard, this paper investigated the effects of elevated temperature on the heterogeneity and anisotropy of SIC norite based on CT scanning and image processing methods. The thermal-induced cracks were reconstructed and analyzed. In addition, the heterogeneity and anisotropy properties of norite were quantified based on a 3-dimensional reconstruction method. The experimental method and study provided a significant reference for the characterization of microstructure and micro-cracks in SIC norite under high-temperature conditions.

## 2. Specimen and Laboratory Test

### 2.1. Specimen

Norite tested in this study was sampled from a Ni ore in Sudbury, Ontario, Canada. The 1.85 Ga old Sudbury structure is an elliptical structure formed by explosive volcanism combined with mafic intrusion [15]. In addition, an impact in such an area was suggested during orogenesis, and such impact could be the origin of REEs in SIC norite [14]. Norite cores from this study were taken 1985 m below the surface of the mining hole collar, with an average diameter of 47.6 mm. SIC norite (hereinafter will be called norite) consists mainly of quartz, feldspar, and biotite which together form an interlock structure. To conduct the CT scanning, cubic specimens were carefully cut from the cores, with approximately 10 mm on each side. The integrity and homogeneity of the rock samples were carefully examined before the thermal treatment.

### 2.2. Thermal Treatment

The prepared norite cubes were heated at a heating rate of 2.5 °C/min in a high-temperature furnace, with an accuracy of ±1 °C. Thermal treatments were classified into four different groups (200 °C, 400 °C, 600 °C, and 800 °C). The slow heat rate was adapted from past literature [1] to avoid thermal shock and generate a homogeneous thermal field in norite. The heated norite specimens were then kept at the condition temperature for 16 h. Afterward, they were cooled down in the furnace before being moved to room temperature.

### 2.3. CT Scan

The CT scanning was conducted by a sub-micron CT scanner located at the multi-scale additive manufacturing (MSAM) lab, University of Waterloo. The scanner includes an X-ray launcher, a specimen platform, and a flat panel detector that receives the X-rays

passing through the specimen, see Figure 1. After scanning, three cross-sections in the directions of *x*, *y* and *z* were collected and analyzed.

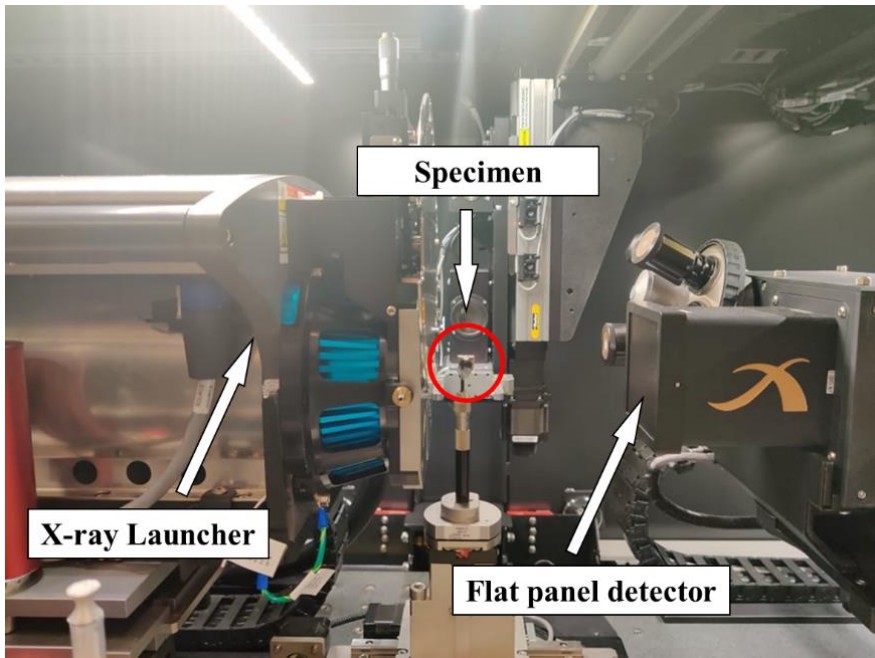

**Figure 1.** CT Scanning set-up for norite after thermal treatment.

*2.4. Image Processing Method*

After acquisitions of CT images, they were preprocessed and analyzed. The major procedure includes image acquisition, image enhancement, shadow elimination, and crack characterization. MATLAB together with ImageJ were used for the image processing. Figure 2 illustrates the general procedure for each specimen. Important steps are introduced respectively in the following sections.

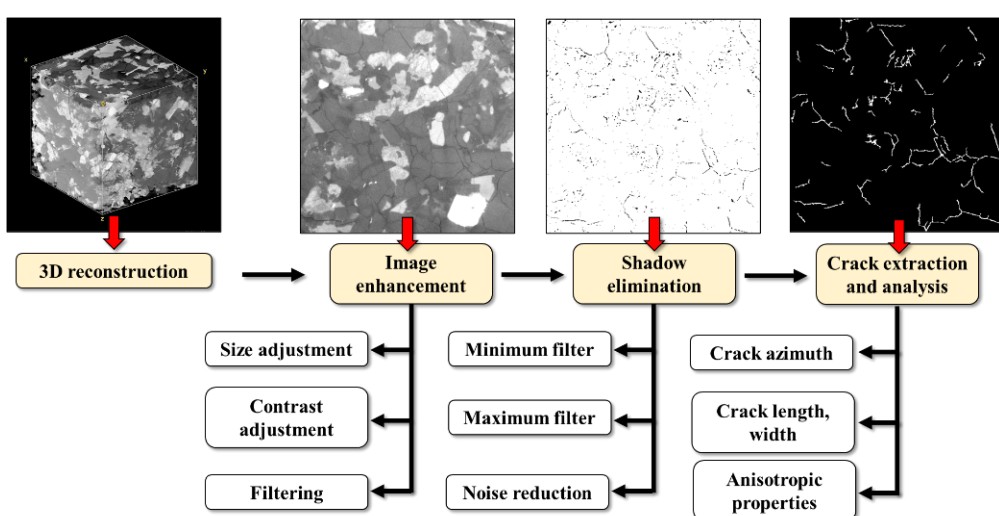

**Figure 2.** Scheme for the quantitative image processing for CT images.

2.4.1. Three-Dimensional Reconstruction

The scanned images were cropped to have the same size (650 × 650 pixels); afterward, a 3-dimensional cube was reconstructed by image stacks. Concerning the scanning setup, one pixel represented 10.02 μm or approximately 10 μm. Therefore, the analyzing cube had a size of 6.5 mm on each side. Figure 3 presents an example of a 3D reconstructed

norite cube and its orthogonal sides. Each cube has three observation planes: *XY* plane (original scanning plane, parallel to ground level and perpendicular to the *z*-axis), *XZ* plane (perpendicular to the *y*-axis), and *YZ* plane (perpendicular to the *x*-axis). It should be noted that the coordinate system and the *x*, *y*, and *z* directions are consistent for all the specimens.

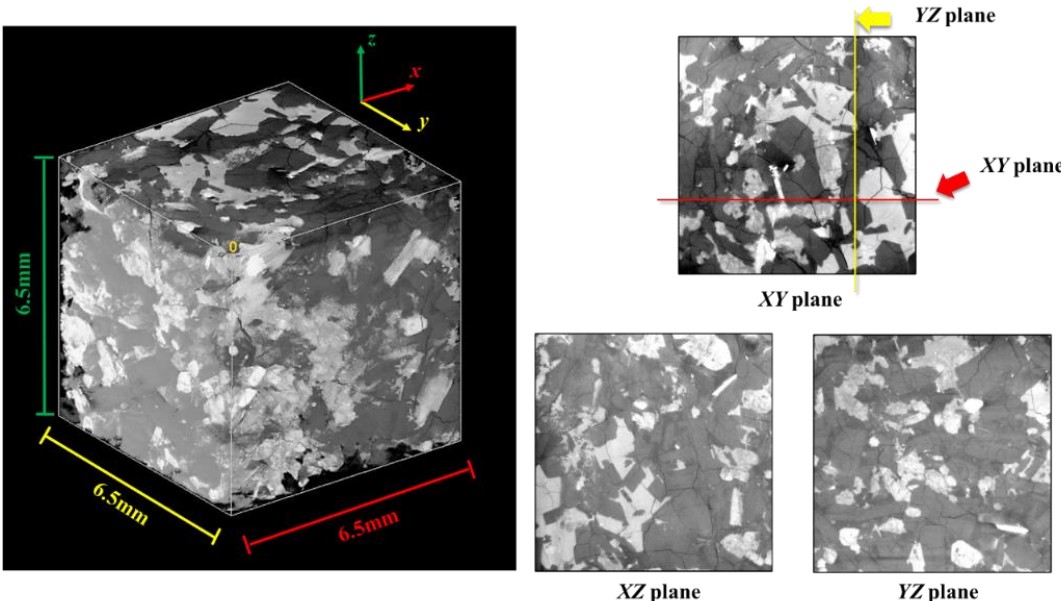

**Figure 3.** Three-dimensional reconstructed norite and its orthogonal slices.

2.4.2. Maximum–Minimum Shadow Filtering (MMSF)

The most used method for image enhancement includes contrast, brightness, and sharpness adjustment. Such adjustment enables further segmentation and crack extraction [16]. However, since the grayscale levels for cracks and some dark minerals are similar, such conventional adjustments only highlighted the cracks but also included the dark minerals and their boundaries with quartz, see Figure 4b).

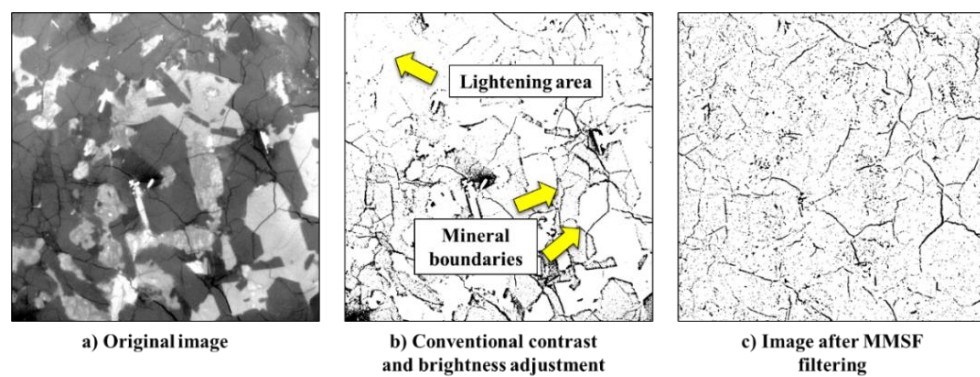

**Figure 4.** Comparisons between (**a**) original image, (**b**) conventional image enhancement, and (**c**) shadow elimination (MMSF filtering).

Therefore, a new filtering method was developed in this study called maximum–minimum shadow filtering (MMSF). The original image was called "O". First, for each pixel (*x*, *y*), the algorithm searched its surrounding area (N × N pixels) and found the maximum greyscale level, P. The pixel (*x*, *y*) was rewritten as P. The returned matrix, A, was called the maximum filter image. Based on A, a similar method was conducted, whereas this time, the algorithm picked the minimum greyscale level, Q, from each pixel surrounding area. Therefore, A was transferred to B, which is called the maximum–minimum filter

image. To have the final filtered image, I, the original image was subtracted by the filtered image, B. See the equation below for details.

$$I = O - B \qquad (1)$$

Figure 4c presents the image after MMSF, it could be seen that the effects of mineral boundaries and the dark connected domain were eliminated. Instead, the image only presents linear-shaped cracks and some scattered noises. Such scattered noises will be removed in the next step. However, it should be noted that the filtering also eliminated some minor cracks and broke some thin cracks into segments. Those cracks will be reconnected and restored in the next step.

### 2.4.3. Crack Extraction and Analysis

During the previous steps, the contrast and sharpness of the image were adjusted, thus generating scattered noise. In addition, cracks with less width might be broken into disconnected segments. In this regard, some image restoration process was conducted, this included, see Figure 5:

(1) Image binarization and invert. Use adaptive thresholding to binarize the image and highlight the thermal cracks in white;

(2) Image close. It performed the morphological closing on the binary image using the structuring element, SE. SE is a linear element used to connect broken segments with a length of three pixels;

(3) Bridge of unconnected segments. Automatically connect the broken segments with infinite calculations until the image became stable;

(4) Small objects delete and minor branch prune. Clear the noise and cut the minor branches generated during the previous image restore steps. Figure 5 illustrates the details of the image restoration process.

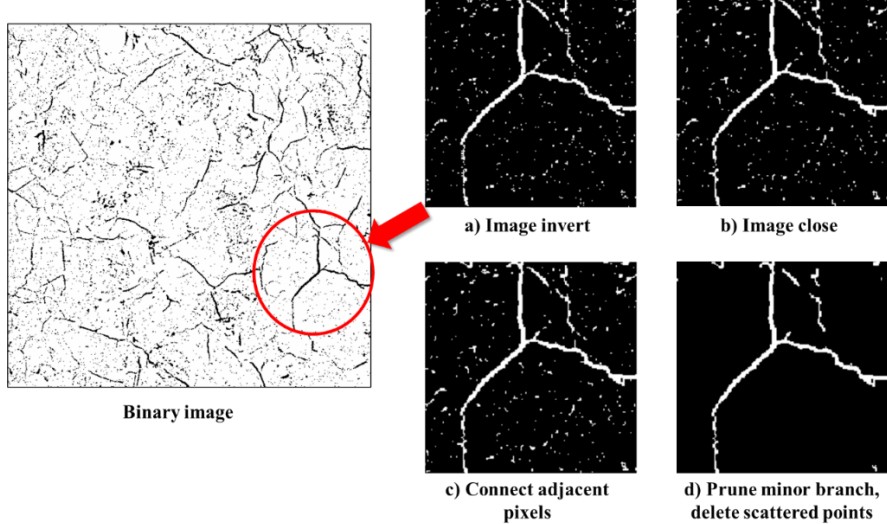

**Figure 5.** Adjustment of MMSF image, prior to crack extraction. Steps include (**a**) image convert; (**b**) image close; (**c**) scattered pixels connection; and (**d**) prune cut.

Finally, the characteristics of each crack were evaluated using the "imageRegionAnalyzer" toolbox in MATLAB. It should be noted that the final output of length and width are in the unit of pixels. Then, they were transferred into μm (1 pixel = 10 μm from the image). The length was defined as the maximum spine length of each crack. On the other hand, orientation was defined as the angle between the line connecting the start and end point of each crack, and the horizontal axis in each image. Orientation ranges between 0° and 360°.

## 3. Results and Discussion

### 3.1. Structure of Norite

Figure 6 presents the structure of the norite after a 200 °C and 850 °C treatment, respectively. Based on different greyscales, the norite was separated into several sub-regions: biotite grains (lighter region), combined quartz and feldspar (dark grey region), and micro-cracks (linear-shaped dark grey and black region). Similar mineral types and identifications were addressed in past studies for Sudbury norite and granite [10,13]. In addition, metal minerals such as Ni were frequently observed in CT scan images. This intermediate-sized mineral has a pure white color and a clear boundary with other minerals. The introduction of Ni and Cu in SIC norites was also documented and presented in past literature [14].

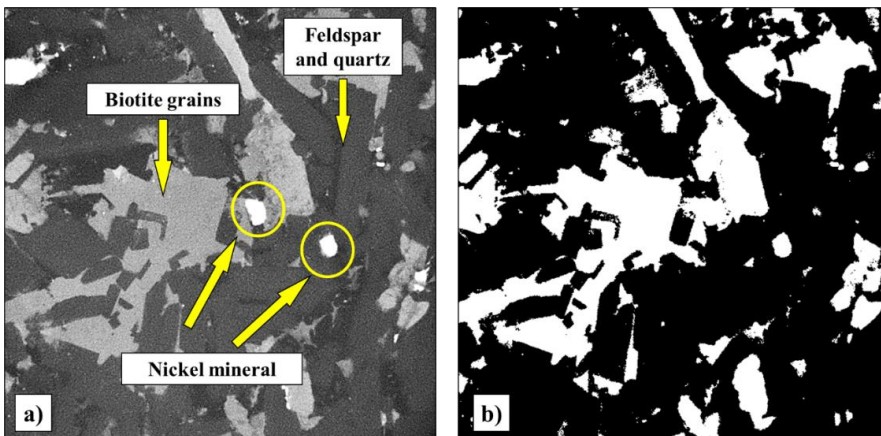

**Figure 6.** Norite after (**a**) 200 °C treatment and (**b**) binary image, at *XY* plane.

To quantify the percentage of minerals, CT images were binarized using the threshold obtained from the Otsu algorithm. The "white" area indicates the biotite grain and the Ni, whereas the "black" area denotes the feldspar and the quartz. In particular, Figure 7 summarizes the percentage of the three mineral categories, based on pixels counting.

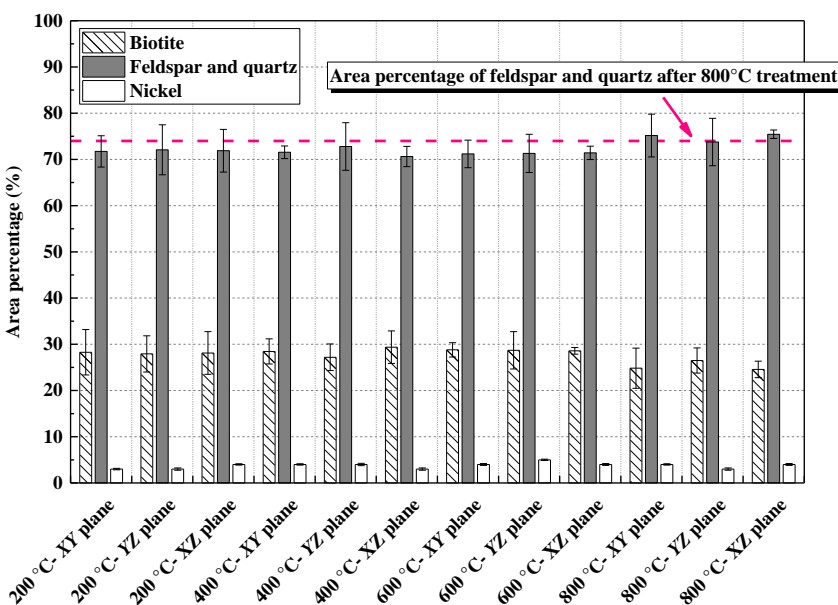

**Figure 7.** Area percentage of different minerals.

It can be seen that the area percentage of biotite ranged between 22% and 27%. For felspar and quartz, they made up between 70% and 75%. In addition, metal minerals (Ni)

occupied approximately 1% to 3% of the total area; however, this might take up more than 5% in some images. Such ratios for each mineral in SIC norite corresponded with the results by physical characterizations from past studies [14,17].

Overall, the percentage for each mineral category in different directions remained consistent. In addition, feldspar and quartz pixel counts in 600 °C- and 800 °C-treated norite were slightly larger than those in lower temperature-treated norite. The reason may be due to the increase in thermal cracks which were showing a similar color compared with dark minerals.

### 3.2. Effects of Thermal Treatment

Figure 8 presents the CT scanning images for SIC norite under 10.0 μm resolution. According to the images, norite subjected to a temperature equal to and under 200 °C showed no significant micro-cracking. With the elevation of temperature (>400 °C), small-scale micro-cracks started to emerge. Such thermal cracks were first observed in the middle of biotite grains (see Figure 9a) with sparse micro-cracks generated due to differences in thermal expansion rates [10]. Further increase in temperature would lead to more micro-cracks in not only biotite but also in quartz and feldspar. The current micro-cracks would also propagate into the adjacent minerals. In addition, thermal-induced tension also generated cracks between different minerals.

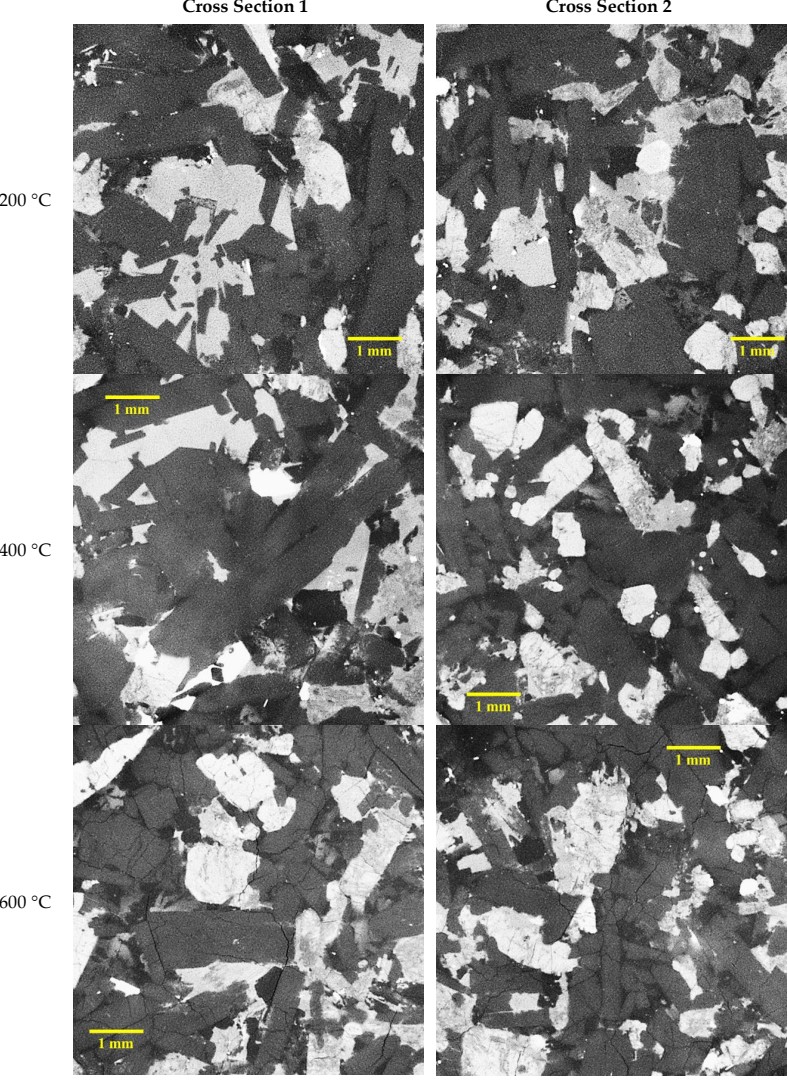

**Figure 8.** *Cont.*

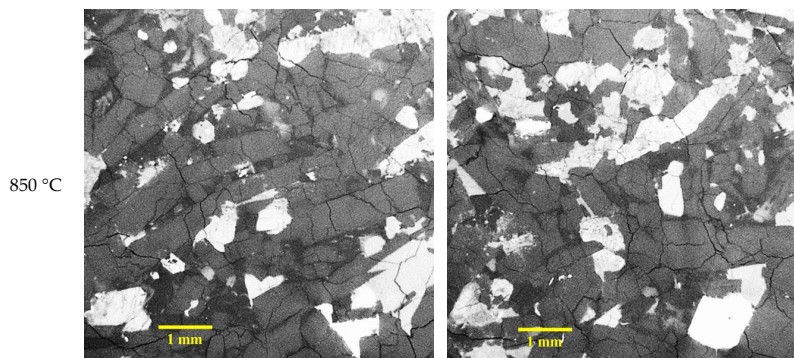

**Figure 8.** CT images of high temperature treated norite, *XY* plane (parallel to ground surface).

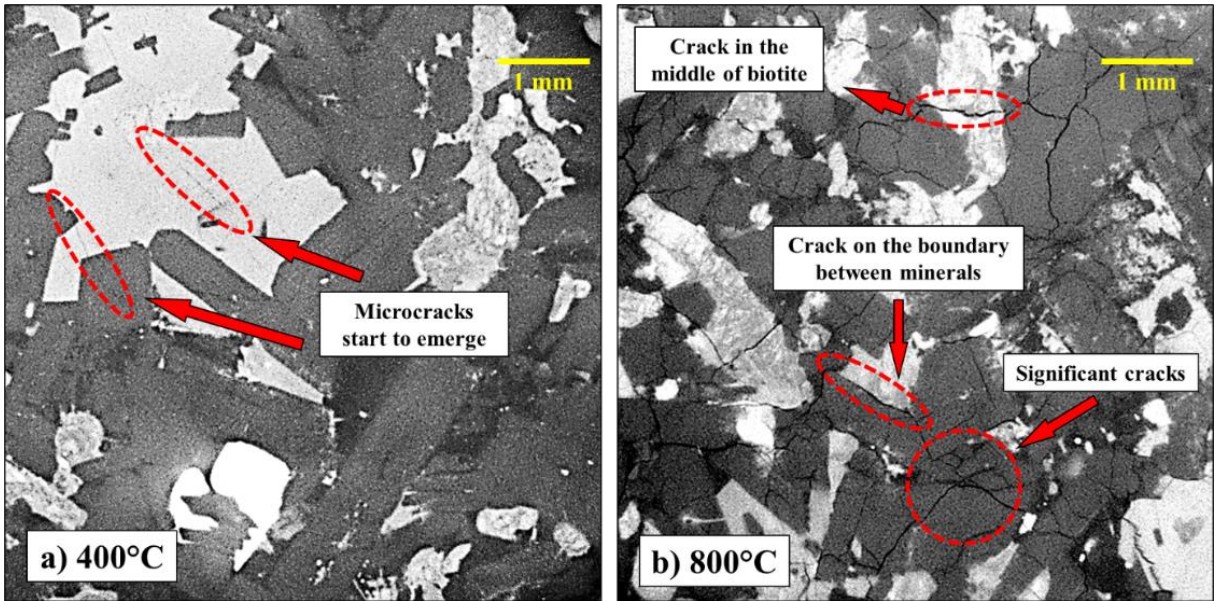

**Figure 9.** Details of thermal cracks on norite after (**a**) 400 °C and (**b**) 800° C treatment.

Visual observation indicated that the length and width of thermal cracks had increased compared to those of the specimens preheated by 200 °C and 400 °C. With the preheating temperature further developed to 800 °C, different micro-cracks started to connect, thus dividing the integral mineral into smaller segments. Figure 9b presents this morphology. Such facts in SIC norites generally correlate with the phenomenon introduced previously in granite, sandstone, and shale rocks [1,6,8]. Nevertheless, the decomposition of carbonate and the clay minerals had not been seen in CT images.

### 3.3. Quantitative Analysis of Micro-Cracks

The following figures (Figures 10–12) summarize the distribution of length, width, and orientations for micro-cracks. Each specimen has three planes (*XY*, *YZ*, and *XZ*). For each plane, five different images were selected from different locations and were analyzed. It should be noted that the quantification for norite after 200 °C was not successful since the micro-cracks were not visible. Objects with a maximum length of fewer than 15 pixels or 150 μm were not taken into account to eliminate the effects of scattered small objects.

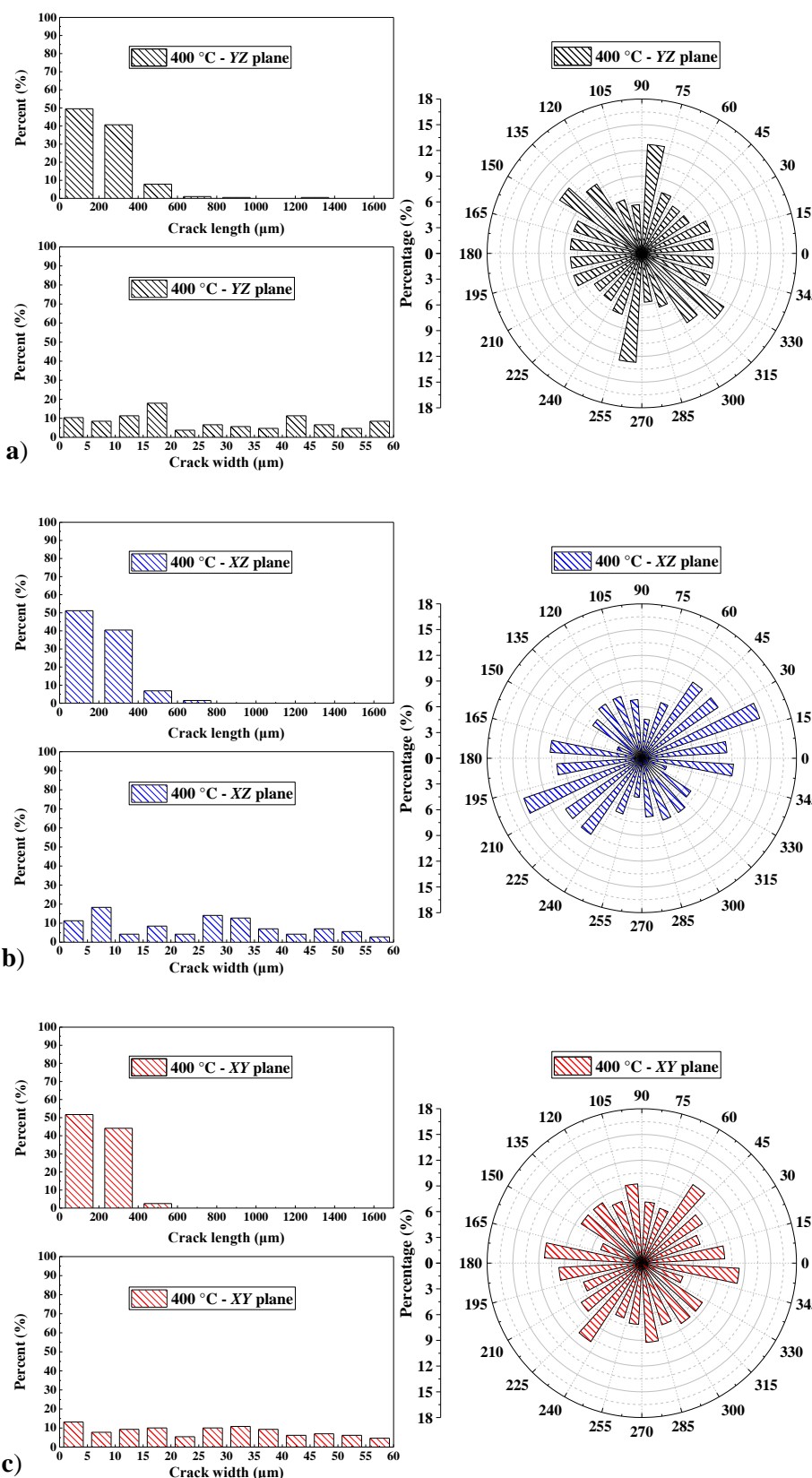

**Figure 10.** Characteristics of cracks for norite after 400 °C treatment, in (**a**) *YZ* plane; (**b**) *XZ* plane; and (**c**) *XY* plane.

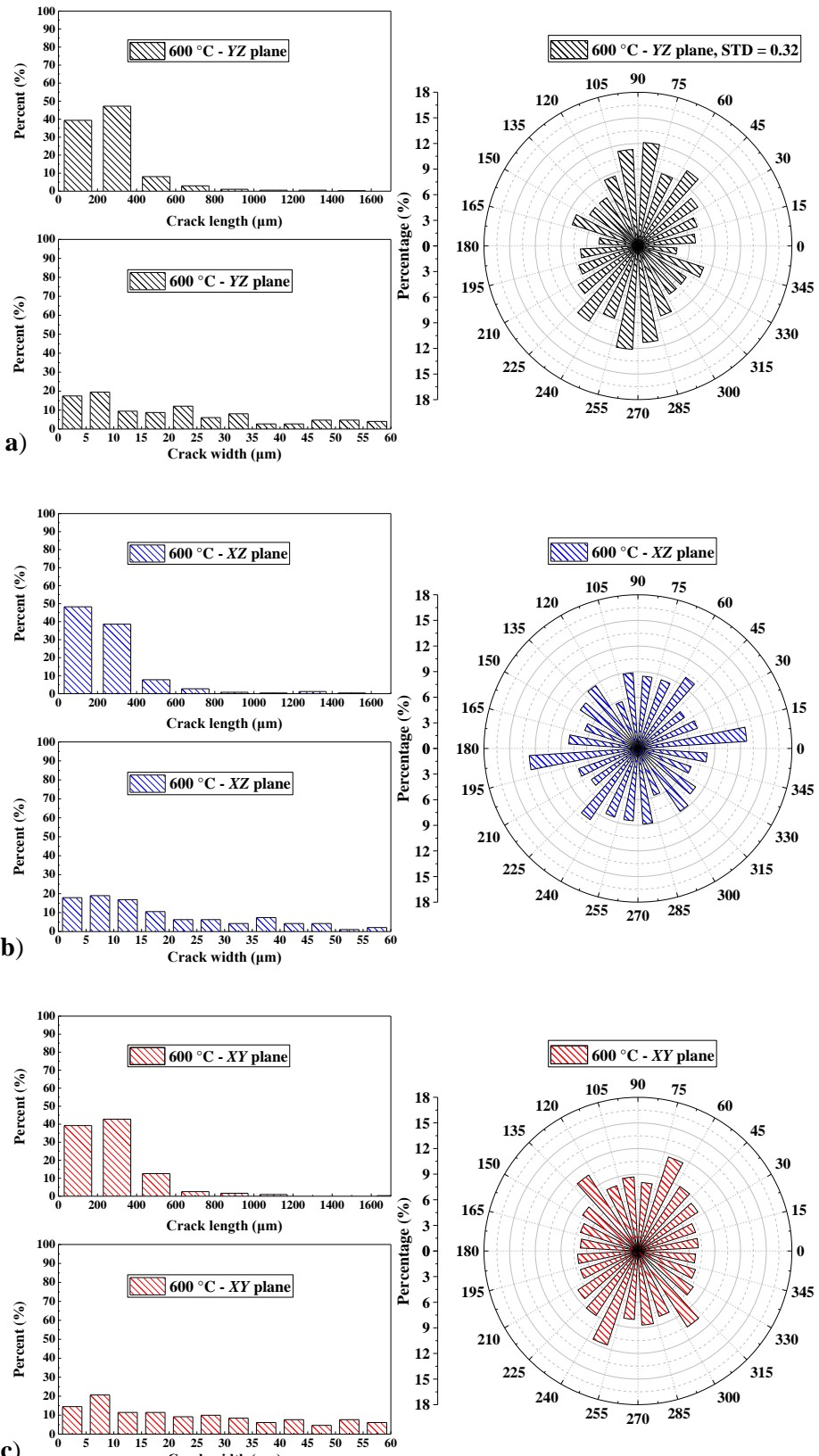

**Figure 11.** Characteristics of cracks for norite after 600 °C treatment, in (**a**) *YZ* plane; (**b**) *XZ* plane; and (**c**) *XY* plane.

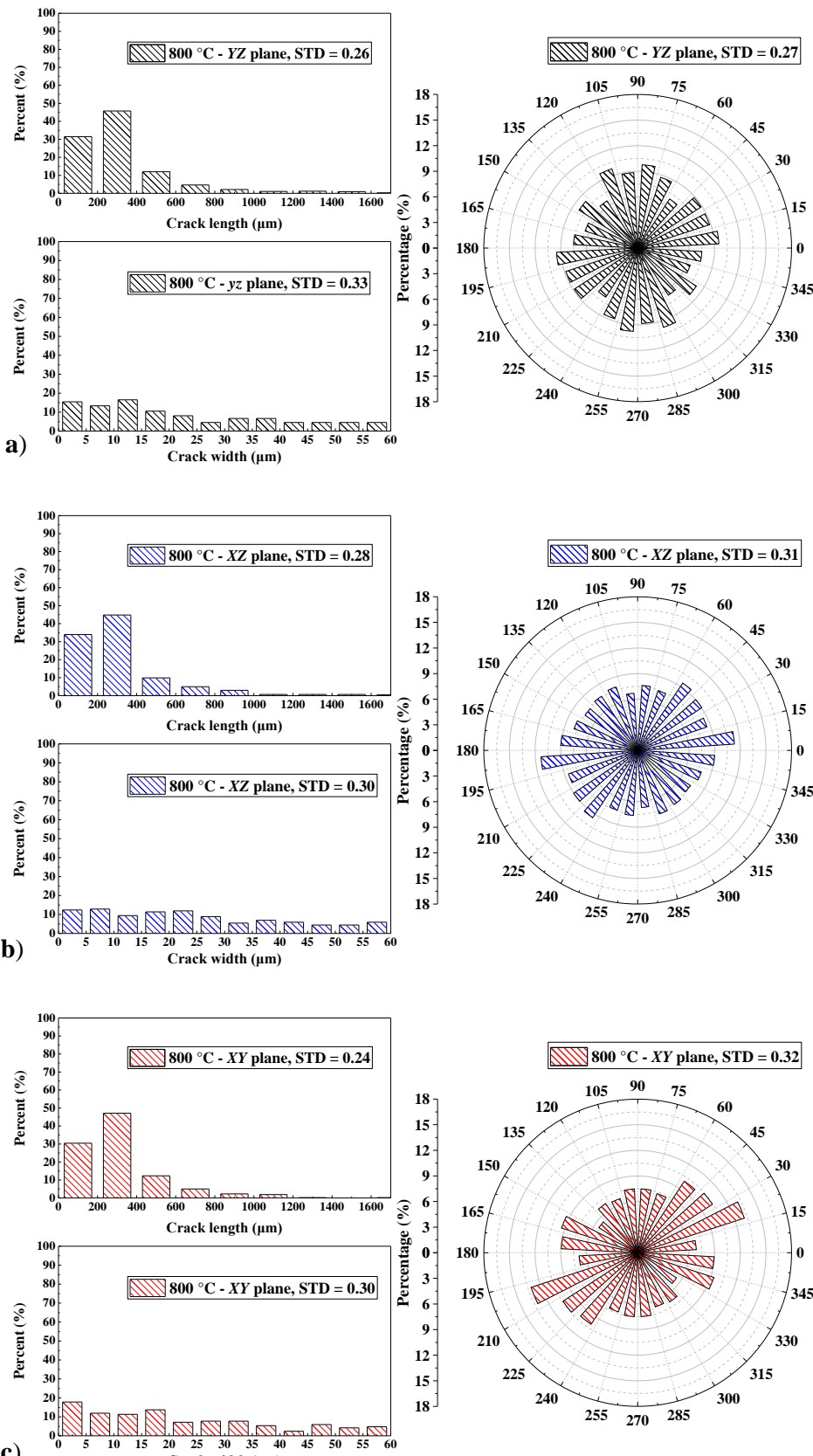

**Figure 12.** Characteristics of cracks for norite after 800 °C treatment, in (**a**) *YZ* plane; (**b**) *XZ* plane; and (**c**) *XY* plane.

The figures indicate that, in 400 °C-treated norite, more than 90% of micro-crack length falls into the two categories (0~200 μm and 200~400 μm). In addition, 5% to 10% of the micro-cracks have a length making up between 400 and 600 μm. On the other hand, the cracks are more likely to have their width ranging between 0 and 15 μm; such micro-cracks are very thin in width and could not be seen by the naked eye. One could also observe the resemblance of crack length and width in three different planes. Thus, it indicates that there were no significant anisotropic properties in 400 °C-treated norite.

With the heating temperature further rising to 600 °C and 800 °C, one of the most distinct facts is the extension of thermal-induced cracks. Such effect was due to the significant volume expansion in quartz minerals associated with the $\alpha$-to-$\beta$ transition at the temperature of 576 °C [9,10]. In 600 °C-treated norite, 200 μm to 400 μm long cracks became the most proportioned thermal-induced cracks. In addition, the increased percentages of longer cracks (crack length ranges between 400 μm and 1200 μm) were also notable. Such a trend was due to the tension increase based on further elevated heating.

On the right side of each figure, the overall orientation of each micro-crack was summarized. The consistency of the crack orientation could be seen from each plane along with different heating temperatures. For example, the most proportioned crack orientation in the *XZ* plane was between 15° and 30° after 400 °C. This range changed slightly to 15° and 30° for both 600 °C- and 800 °C-treated norites. The trend, the growth and propagation of cracks going in the same direction, could be due to the effects of the deformation twin [18]. Another notable phenomenon is the even distribution trend of crack orientations with higher temperature-treated norites, which is particularly noticeable from the *YZ* planes. The reason for this could be that, with a higher temperature treatment, more cracks propagate in the norite. In a generally homogeneous structure, the minor micro-cracks would occur in any direction, therefore, adjusting the overall distribution of crack orientations. Similar findings were reported by the research on the heterogeneity properties of heated granite [1]. It should be noted that the development of crack length and orientations could be more evident by performing continuous CT scanning on specimens under temperature growth. Moreover, the pretreatment of CT images and the image processing method, especially MMSF filtering, might slightly change the overall results on crack distributions. The effects of such methodology and further validations could be investigated by introducing more types of rocks in the future.

### 3.4. Anisotropic Analysis of Treated SIC Norite

The areal porosity, $D_{i,n}$, of high temperature-treated SIC norite was defined in the following Equation (2). It should be noted that once the first surface $D_{z,1}$ was calculated, then $D_{x,1}$ and $D_{y,1}$ were acquired from their corresponding orthogonal slices, as is presented in Figure 3. Quantification of $D_{i,n}$ was conducted by using the "bwarea" function in MATLAB. The results of $D_{i,n}$ are summarized in Figure 13.

$$D_{i,n} = \frac{S_{i,n}}{S_{total}} \times 100\% \tag{2}$$

where

$S_{i,n}$ = area of cracks and voids;
$S_{i,n}$ = total area of each image;
$i$ = direction of each plane = $x$ (*YZ* plane), $y$ (*XZ* plane), $z$ (*XY* plane);
$n$ = number of each image at the same direction = 1, 2, 3, . . .

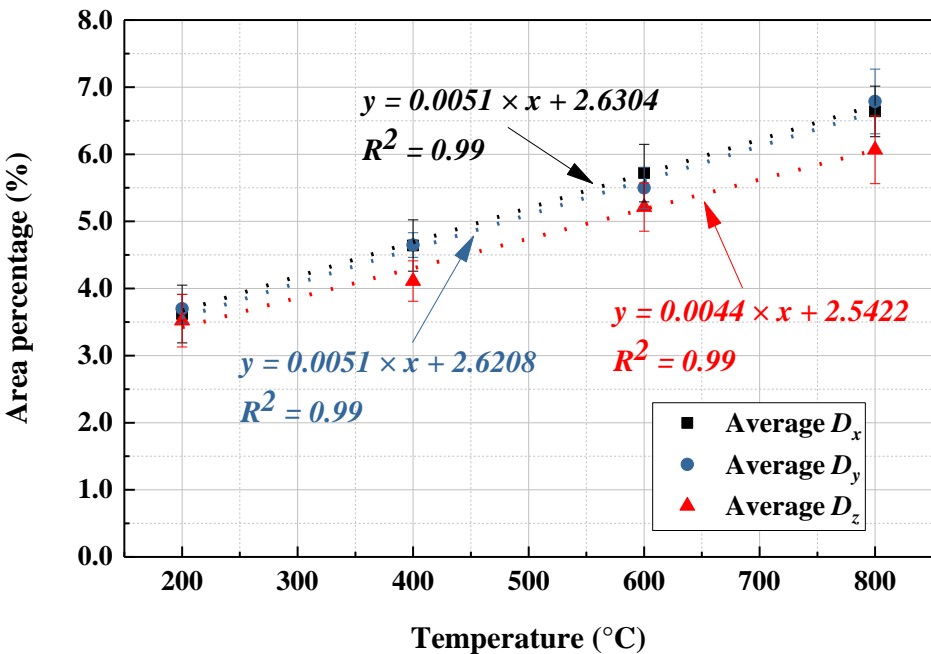

**Figure 13.** Average porosity of SIC norite in each direction.

Not surprisingly, an increase in porosity was observed with the increase in treatment temperature, indicating the growth of cracks and voids. From this study, however, a linear relationship between average areal porosity $D_{i,n}$ and the treatment temperature was shown, with a high coefficient of determination values. This linear relationship was also reported in past literature, especially when the temperature rose from 400 °C to 1000 °C [8,10]. In addition, the rate and porosity of $D_x$ and $D_y$ were very similar to each other, indicating a homogeneous structure in the horizontal plane. On the other hand, $D_z$ was a little far from $D_x$ and $D_y$, denoting a slight difference in voids and cracks between vertical and horizontal directions.

Furthermore, the anisotropy coefficient, $A$, is introduced to evaluate the anisotropy of SIC norite after thermal treatment. The definition of $A$ for each direction was based on Equation (3) [1]. Results were plotted in Figure 14.

$$A_{x/z} = \frac{|D_{x,n} - D_{z,n}|}{D_{z,n}}; \ A_{y/z} = \frac{|D_{y,n} - D_{z,n}|}{D_{z,n}}; \ A_{x/y} = \frac{|D_{x,n} - D_{y,n}|}{D_{y,n}} \tag{3}$$

where $D_{x,n}$, $D_{y,n}$, $D_{z,n}$ = areal porosity at the slice of $D_{z,n}$, and its corresponding $D_{x,n}$ and $D_{y,n}$; $A_{x/z}$ = anisotropy coefficient for $x$-direction compared with $z$-direction.

From 200 °C to 800 °C, the anisotropy coefficient for all the slices ranged between 0% and 20%. In general, the anisotropy coefficients $D_{x,n}$ and $D_{y,n}$ peaked at around 16% at the temperature of 400 °C, then they decreased slightly to between 10% and 15%, at the temperature of 600 °C and 800 °C. Such a trend was also reported recently [1]. The reason may be due to the intensive generation of cracks between 400 °C and 600 °C. However, with further temperature increases, the $D_{x,n}$ and $D_{y,n}$ remained constant. It should also be noted that, at high temperatures (>600 °C), different slices have more scattered anisotropy coefficient values. On the other hand, $D_{x,y}$ remained constant throughout the temperature change. Overall, the anisotropic properties of SIC norite were noticeable between the vertical and horizontal directions. However, within the horizontal plane, the structure was more homogeneous (between *XZ* plane and *YZ* plane).

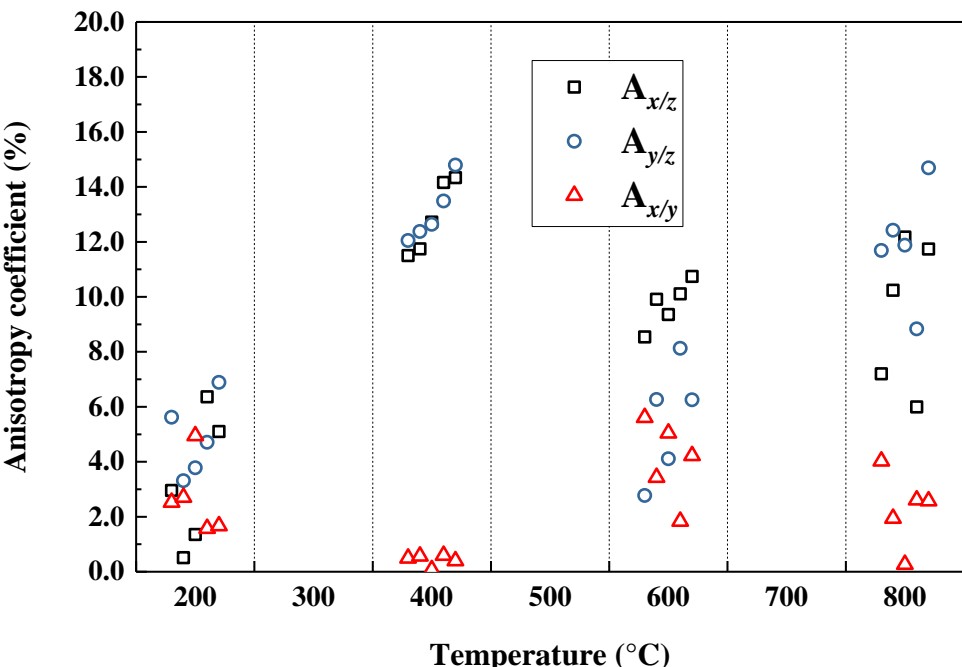

**Figure 14.** Anisotropic coefficients of SIC norite after treatment.

It was indicated from the literature that the orientation of micro-cracks in rocks would significantly influence the anisotropic properties of strength and modulus [19–21]. The longer the crack, the more important effect it may have on the formation and development of the failure pattern of rocks. During these studies, cracks that have lengths of around 500 μm (0.5 mm) were found to have the highest frequency. Therefore, the following Figure 15 presents the orientations of "major micro-cracks" (crack length longer than 500 μm) after 600 °C and 800 °C. It should be noted that the number of major micro-cracks generated at lower temperatures was too low, so these "short" length cracks were not analyzed.

As Figure 15 indicates, the orientations of major micro-cracks in the *XY* plane and *XZ* plane were generally consistent between 600 °C and 800 °C. Their most proportioned crack orientations accounted for between 0° and 60° and 120° and 195°, respectively. Such orientation ranges were slightly different from those illustrated in Figures 11 and 12. On the other hand, the distribution of orientations in the *XZ* plane became greater at 800 °C, and such a trend aligned with the situations in Figures 11 and 12.

In general, SIC norite did not show significant anisotropic properties with the increase in treatment temperature. However, several orientations have higher frequencies for thermal-induced micro-cracks, especially in *XY* and the *XZ* planes thus could further influence the damage pattern of norite.

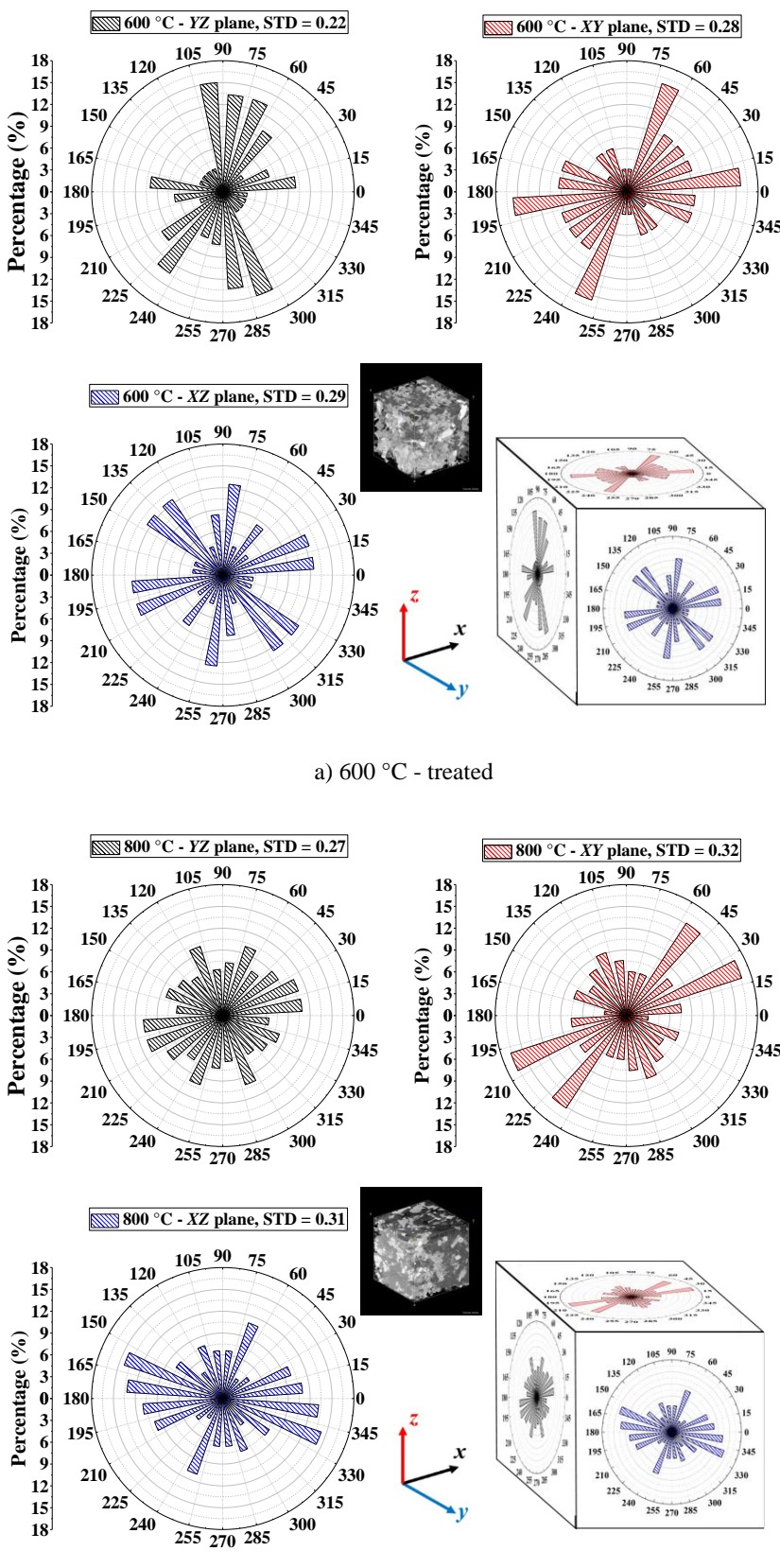

**Figure 15.** Orientations of major micro-cracks in (**a**) 600 °C and (**b**) 800 °C-treated SIC norite.

### 4. Conclusions

Based on the laboratory tests and analysis of the data, several specific conclusions and discussions can be addressed, as follows:

(1) The areal percentage of biotite ranged between 22% and 27%. The rest was occupied with felspar, quartz, and a small amount of metal minerals. Small-scale micro-cracking was first observed in the middle of biotite grains at the temperature of 400 °C. A further increase in temperature would generate more micro-cracks not only in biotite but also in quartz and feldspar. With the preheating temperature further developed to 800 °C, different micro-cracks started to connect, thus dividing the integral mineral into smaller segments;

(2) The image processing method together with the MMSF introduced in this study clearly identified the thermal-induced cracks. Quantitative analysis of pre-treated images indicated that most micro-crack lengths fell into the two categories (0~200 μm and 200~400 μm). On the other hand, the overall length of micro-cracks developed with the increase in preheating temperature, especially when it reached and exceeded 600 °C;

(3) The orientation of cracks exhibited two major trends with the increase in temperature: they followed a similar distribution pattern (in the *XZ* and *XY* planes) or became more evenly distributed (in the *YZ* plane);

(4) A linear relationship between the average porosity of SIC norite and the treatment temperature was found in each direction. Such relationships for the *YZ* and *XZ* planes were very similar, with a slight difference from plane *XY*. This indicated the anisotropic properties between vertical and horizontal directions of norite. This fact was further confirmed by the anisotropic coefficients which soared sharply from 200° C to 400 °C and then declined at 600 °C and 800 °C.

Overall, the paper presented a comprehensive study on the effects of thermal treatment and the anisotropic properties of SIC norite. The image processing tools were developed and used for the quantitative study of thermal-induced micro-cracks generated at different temperatures. It should be noted that the MMSF filtering treatment and image processing methods should be further validated by performing an investigation on other types of rocks. In addition, the crack propagation characteristics would be more evident under scanning observations during the heating process. Such limitations could be improved by future microscopic investigations on different rock samples, and by introducing advanced CT scanning equipment. Nevertheless, the methodology and findings from this paper will be an important reference for future studies regarding the thermal effects on norite and similar rocks.

**Author Contributions:** Conceptualization, S.-L.W. and S.Y.; methodology, S.-L.W. and S.Y.; software, S.-L.W. and J.H.; validation, S.Y. and B.S.; formal analysis, S.-L.W.; investigation, S.-L.W.; resources, B.S.; data curation, S.-L.W.; writing—original draft preparation, S.-L.W.; writing—review and editing, S.Y. and J.H.; visualization, S.-L.W.; supervision, S.Y.; project administration, S.Y. and B.S. All authors have read and agreed to the published version of the manuscript.

**Funding:** This research received no external funding.

**Data Availability Statement:** Data are contained within the article. MATLAB and ImageJ codes are available on reasonable request from the corresponding author.

**Acknowledgments:** This research received no specific grant from any funding agency from the public, commercial, or not-for-profit sectors.

**Conflicts of Interest:** The authors declare no conflict of interest.

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
