# Peer review of "An Investigation of Thermal Effects on Micro-Properties of Sudbury Norite by CT Scanning and Image Processing Method"

_mining, doi:10.3390/mining2030027_

Round 1

Reviewer 1 Report

This paper is presenting a framework for studying the effect of thermal treatment on crack development in norite. The authors are extracted cracks by applying a novel image processing technique using the so-called MMSF filter. Then, they have presented how the cracks can be analysed and compared under different thermal treatments.

The introduction is good and summarises background, aim and contribution of this research. The methods are well developed and explained in an easy to understand manner. The results are well presented and discussed. Apart from some typos like the following, 1um=10Px (should be 1px = 10um), there is no significant issue throughout the paper. 

My only recommendation is to add limitation of the present work as well as some directions for future work either in conclusion or in a separate section before conclusion under the title of limitation and future work.

All in all, I recommend this paper to be accepted for publication after a minor revision.

Author Response

Authors: Thank you so much for your comments. We have corrected the typos and did carefully proofreading throughout the paper.  In addition, we added discussions in several paragraphs and in conclusion to indicate the current limitations of this research and the future directions (Line 277-282; 386-391). They include the validation of MMSF filtering method on different rock types and the continuous scanning for norite under the temperature growing. Such limitations could be improved by future investigations on different rock samples, and by introducing advanced CT scanning equipment.

Reviewer 2 Report

The article “An investigation of thermal effects on micro-properties of Sud-2 bury norite by CT scanning and image processing method” is interesting and contemporary. The authors have provided a commendable study. There are some issues with the study which I have highlighted in the attached pdf file. The authors should address all the queries in the attached file.

There are some major issues with the study;

1. Error! Reference source not found (this error should be resolved);

2. Captions are written properly for most of the figures. Mention all the sub-figures in the caption such as a), b), c), etc.;

3. Update the literature;

4. Conclusions should be concise and less.

Author Response

  1. Error! Reference source not found (this error should be resolved);

Authors: Such error was caused by cross-references. The issue has been resolved.

  1. Captions are written properly for most of the figures. Mention all the sub-figures in the caption such as a), b), c), etc.;

Authors: Captions have been revised as suggested.

  1. Update the literature;

Authors: The list of literature has been updated as suggested.

  1. Conclusions should be concise and less.

Authors: The conclusions have been rearranged as 4 bullet points. On the other hand, the last paragraph summarizes the overall results and discusses the limits and future research direction. Therefore, is not in the same level as the bullet points.

Round 2

Reviewer 2 Report

The authors have responded to all of my queries and acted accordingly as per my suggestions and recommendations. The revised manuscript is an improved version in all aspects and I recommend it for publication.